# Exercise Adherence in Men with Prostate Cancer Undergoing Androgen Deprivation Therapy: A Systematic Review and Meta-Analysis

**DOI:** 10.3390/cancers14102452

**Published:** 2022-05-16

**Authors:** Kellie Toohey, Maddison Hunter, Catherine Paterson, Reza Mortazavi, Benjamin Singh

**Affiliations:** 1Faculty of Health, University of Canberra, Bruce, ACT 2617, Australia; maddy.hunter@canberra.edu.au (M.H.); catherine.paterson@canberra.edu.au (C.P.); reza.mortazavi@canberra.edu.au (R.M.); 2Prehabilitation, Activity, Cancer, Exercise and Survivorship (PACES) Research Group, University of Canberra, Bruce, ACT 2617, Australia; 3School of Nursing, Midwifery and Paramedic Practice, Robert Gordon University, Aberdeen AB10 7QB, UK; 4Allied Health and Human Performance, University of South Australia City East Campus, Adelaide, SA 5001, Australia; benjamin.singh@connect.qut.edu.au

**Keywords:** prostate cancer, androgen deprivation therapy, exercise, adherence

## Abstract

**Simple Summary:**

Prostate cancer treatments, including androgen deprivation therapy, can lead to a range of undesirable physical and psychological alterations for men. Participating in regular exercise has been shown to reduce the severity of these changes, providing an opportunity to improve the lives of these patients. There are a range of exercise interventions described in the literature; however, it is unknown what the optimal type of exercise to encourage adherence is. This systematic review and meta-analysis investigates exercise intervention adherence of patients receiving androgen deprivation therapy while identifying some of the effects of exercise on some physiological outcomes. It also includes a qualitative perspective to describe the issues relating to exercise for this population in both real-life and intervention settings. This research is vital, as future research may benefit from the understanding of the factors that will encourage exercise participation in this population.

**Abstract:**

Androgen deprivation therapy (ADT) for prostate cancer treatment is associated with adverse physiological changes; however, exercise can improve outcomes. This systematic review and meta-analysis aimed to determine exercise intervention adherence and its effects on physiological outcomes in men diagnosed with prostate cancer undergoing ADT. Uniquely, this review incorporated a meta-aggregation of qualitative data, providing perspectives from the men’s experiences. A systematic review and meta-analysis were completed following PRISMA guidelines. Databases (CINAHL, Cochrane, PubMed) were searched for studies using “prostate cancer”, “exercise intervention”, and “androgen deprivation therapy”. Quantitative randomised controlled trials describing adherence to exercise interventions were selected, with qualitative articles selected based on descriptions of experiences around participation. Subgroup meta-analyses of adherence, exercise mode, and intervention duration were completed for quality of life, aerobic fitness, fatigue, and strength. In total, 644 articles were identified, with 29 (*n* = 23 quantitative; *n* = 6 qualitative) articles from 25 studies included. Exercise had no effects (*p* < 0.05) on quality of life and fatigue. Significant effects (all *p* < 0.05) were observed for aerobic fitness, and upper- and lower-body strength. Adherence to exercise-based interventions was 80.38%, with improvements observed in aerobic fitness and strength. Subgroup analysis revealed exercise adherence impacted fatigue and strength, with greater improvements observed in programs >12-weeks.

## 1. Introduction

In developed countries, prostate cancer is one of the most prevalent cancers in men, accounting for one in five new cancer diagnoses [1]. Prostate cancer risk factors include non-modifiable factors of older age, family history, and ethnicity [2]. Additionally, smoking and obesity have been identified as some of the modifiable risk factors for disease development [2]. The survival rate for prostate cancer is 98% [1] meaning that the number of men requiring rehabilitation to address survivorship needs is at an all-time high.

Prostate cancer is hormone dependent, with the androgens testosterone and dihydrotestosterone (DHT) responsible for driving disease progression [3]. One of the most common treatment modalities for prostate cancer is androgen deprivation therapy (ADT) [4], with approximately half of patients receiving this therapy over their courses of treatment, either as a primary or adjuvant therapy [5]. The purpose of ADT is to reduce the blood levels of androgens through either surgical or medical interventions, including androgen-targeted therapy [4].

However, the ADT-dependent reduction in testosterone levels can lead to various side effects, many of which are attributed to alterations in metabolism [6]. These side effects include the significant risk of the development of obesity, diabetes mellitus, and cardiovascular disease [6], with an inverse association observed between serum testosterone levels and cardiovascular disease risk [7]. Further side effects include decreased libido, hot flashes, reduced sexual function, impaired quality of life, and altered psychosocial well-being [8]. Patients also experience a change in physical health, including reduced muscle strength and altered body composition [9].

There is substantial support, including by Exercise and Sports Science Australia (ESSA), for the participation in exercise for patients with cancer, due to the multisystem benefits experienced [10]. These benefits comprise of improvements in physical and psychological function, quality of life, and overall well-being [11].

Substantial research has been completed investigating the positive outcomes of exercise in improving adverse ADT side effects [12,13,14,15]. However, to the best of our knowledge, research has not defined the optimal type of exercise intervention or prescription to encourage adherence and attendance by men with prostate cancer. This systematic review and meta-analysis aimed to evaluate the adherence of patients with prostate cancer receiving ADT to exercise interventions, considering the effects of exercise on quality of life, fatigue, aerobic fitness, and muscle strength. Qualitative studies were also included in this review (as a separate analysis) to provide an evidence informed approach to address issues of adherence to exercise programs for men living with prostate cancer.

## 2. Materials and Methods

### 2.1. Search Strategy

This systematic review was completed according to Preferred Reporting Items for Systematic Reviews and Meta-Analyses (PRISMA) guidelines [16]. A search of the electronic databases, CINAHL, Cochrane, and PubMed was completed for manuscripts published between January 2000 to July 2021. Searches were limited to identifying articles involving human participants, that were published in English, and were published in peer-reviewed journals. The reference lists of articles were also searched to identify additional relevant articles. The search strategy terminology included the following title and keyword search terms: “Prostate Cancer OR Prostate Neoplasm” AND “Exercise OR Physical Activity OR Exercise Intervention” AND “Cancer OR Oncology” AND “Androgen Deprivation Therapy”. The review was registered with the International Prospective Register of Systematic Reviews database (PROSPERO: CRD42020190291).

### 2.2. Study Selection

#### Inclusion and Exclusion Criteria

Studies of quantitative, qualitative, or mixed method design published in English in peer-reviewed journals were included in this review. Articles were considered for inclusion if they investigated the adherence to an exercise intervention of adult (>18 years of age) men diagnosed with prostate cancer receiving ADT treatment. Specifically, quantitative studies were required to describe a prescribed exercise intervention, include a control group of usual care involving men with prostate cancer receiving ADT, provide a comparison between pre- and post-intervention, and describe adherence levels to the exercise. Qualitative studies were selected for inclusion if they described the experiences of this population in reference to any type of exercise for both self-driven (i.e., unsupervised) exercise and formal interventions (i.e., supervised), and issues around adherence. Exclusions for this review included reviews of any type, conference abstracts, editorials, clinical trial protocols, cross-sectional studies, and case reports; however, the reference lists of these records were searched for any relevant articles that were missed in the original literature search.

### 2.3. Data Extraction

Following the searches of the electronic databases, articles were imported to the data management software Covidence (v2151, Melbourne, Australia), where duplicates were removed (Figure 1). Two authors (M.H. and C.P.) reviewed the titles and abstracts to identify the articles appropriate for full-text analysis, while a third author (R.M.) resolved any conflicts. A full-text review was then completed by two authors (M.H. and K.M) to assess each article for the inclusion criteria. Data relating to the study characteristics, such as the study population, exercise intervention, and outcomes, including adherence and changes in exercise outcomes compared to baseline, were extracted by all authors. Quantitative articles that did not include (1) a control group, (2) participants who received ADT, (3) physical and/or psychosocial outcomes, or (4) reports of participant adherence were excluded.

### 2.4. Assessment of Study Quality

Methodological quality assessment of the included studies was completed using the Mixed Methods Appraisal Tool (MMAT 2018) [17]. The quality assessment was completed independently by two authors (M.H. and K.M.), with a third author (K.T.) discussing any disagreements. Each included study was assigned a score based on the information provided in the appraisal tool, with a rating of 2 indicating a low risk of bias, 1 indicating an unclear risk of bias, and 0 indicating a high risk of bias.

### 2.5. Data Analysis

Meta-analyses were undertaken to evaluate the effects of exercise on quality of life, aerobic fitness, fatigue, and muscle strength (upper- and lower-body). Outcomes were analysed as continuous variables and involved comparisons of post-intervention means and standard deviations (SD) for the intervention compared with control groups. To allow comparisons of data from different scales, standardised mean differences (SMDs) were used as the effect measure (calculated using RevMan software, version 5). Forest plots for each meta-analysis were created using R software (version 4.4.2). The original authors were contacted, or means and/or SD were calculated using reported data using recommended formulas [18] if means and/or SD were not reported in an article. If multiple methods of assessing an outcome were reported in an article, the method that was the gold standard or with demonstrated validity and reliability was used for the meta-analysis.

Data were pooled at the study level for each meta-analysis. To assess publication bias, a funnel plot was used to graph SMDs and standard errors against each other, and asymmetries and missing sections within the plot were assessed [19]. Cochran’s Q test was used to assess statistical heterogeneity and the proportion of the outcome that was attributed to variability was assessed using the *I*^2^ statistic [20,21] as follows: 0–29%: no heterogeneity; 30–49%: moderate heterogeneity; 50–74%: substantial heterogeneity; 75–100%: considerable heterogeneity [21]. Planned subgroup analyses were performed to evaluate the effects of: (1) adherence to the intervention (>75% adherence, ≤75% adherence and adherence not reported as the number of completed sessions); (2) exercise mode (aerobic-based, resistance-based, mixed mode (combined aerobic- and resistance-based), and other); (3) study duration (12 weeks or less and greater than 12 weeks). The following values were used to classify the magnitude of effects <0.20: a small effect; 0.20–0.50: medium effect; >0.50: a large effect [22]. A *p*-value < 0.05 was considered statistically significant.

## 3. Results

### 3.1. Study Selection

The literature search of the electronic databases (Figure 1) identified 642 articles, with secondary searches of the articles reference lists identifying a further two articles. Following the removal of duplicates (*n* = 228) and irrelevant studies based on an in-depth title and abstract screening (*n* = 316), full-text review analysis was completed for 100 articles. Further article exclusion following the full-text review (*n* = 71) resulted in the inclusion of 29 articles fitting the pre-defined eligibility criteria [23,24,25,26,27,28,29,30,31,32,33,34,35,36,37,38,39,40,41,42,43,44,45,46,47,48,49,50,51]. Of note, three studies (Segal et al. 2003 [26,44]; Focht et al. 2018 [28,29]; Uth et al. 2014 [45,46,47]) involved multiple publications on the same study. Therefore, a total of 29 articles reported from 25 studies were included. The included articles comprised quantitative randomised controlled trials (*n* = 23) [24,25,26,27,28,29,30,31,32,34,37,38,39,40,41,42,44,45,46,47,48,49,50] and a series of qualitative studies (*n* = 6) [23,33,35,36,43,51]. The qualitative studies consisted of focus groups (semi-structured (*n* = 2) [36,43] and otherwise unspecified (*n* = 1) [23]), and semi-structured interviews (*n* = 3) [33,35,51] as methods of data collection.

### 3.2. Study Characteristics

The characteristics of the included studies are included in Table 1 and Table 2. A total of 1321 participants (quantitative study participants *n* = 1223; qualitative study participants *n* = 98) were included. The included participants had heterogeneous clinical characteristics including cancer stage, and ADT duration, including those recently diagnosed and commencing ADT, and those at end stages of prognosis receiving ADT. For the quantitative studies, the sample sizes ranged from 19 to 155, with the average age ranging from 64.5 to 77.54 years, and the sample sizes for the qualitative studies ranged from 3 to 29 and ages ranged from 60 to 88 years.

For the selected quantitative studies, the majority involved exercise programs of combined aerobic and resistance training [24,25,27,28,29,31,34,38,45,46,47,48,49,50], with the remaining studies involving isolated aerobic [30,40,41,42] or resistance [26,32,37,39,44] programs. There was a variety of exercise supervision described for the included studies, ranging from completely supervised exercise programs [25,26,28,29,31,32,39,44,45,46,47] to completely unsupervised [30,37,40,41,42,49], with some studies including combined supervision (tapered supervision) [24,27,34,38,48,50], with both group [24,25,27,28,29,31,32,34,39,45,46,47,50] and individual [26,30,37,38,40,41,42,44,48,49] exercise programs included. A variety of settings were utilised for the included studies, including gyms [24,25,26,28,29,31,32,39,44], at home settings [30,37,40,41,42,49], or combined gym and home [27,34,38,48,50], with one study completing their exercise at a football training venue [45,46,47]. The participants in the selected qualitative studies had experienced a variety of exercise throughout their ADT treatment, including formal exercise interventions [23,35], non-research exercise programs [43], and self-guided exercise [33,36,51].

### 3.3. Quality Appraisal Results

The results of the quality appraisal of the assessed studies are presented in Table 3, where no studies reported a high risk of bias across all domains. All studies reported a low risk of bias for the first two domains, describing the outcomes of the studies addressing the research questions. Similarly, for the quantitative studies, all groups were found to be comparable at baseline, and the reported outcome data were complete. Most of the unclear bias risk was observed for outcome assessors being blinded to the intervention, where the articles did not include a description of blinding [25,26,27,30,31,32,34,37,39,40,45,46,47,48,50]. Furthermore, the only high bias risk observed for the included articles was for the same domain, with one article stating that outcomes were not blinded [41]. Additional issues with the quantitative studies included the unclear observations of participant adherence [31,32,37,45], which is linked to the primary outcomes of this review observing compliance to exercise. Only one study reported an unclear bias, where it was unclear if the qualitative approach was appropriate [33], with the bias for the qualitative studies low overall.

### 3.4. Meta-Analyses

Results of the overall effects of exercise-based interventions on quality of life, aerobic fitness, fatigue, upper-body strength, and lower-body strength are shown in Figure 2. Exercise-based interventions had no overall effects on quality of life (SMD = 0.15, 95% CI = −0.03, 0.32; *p* = 0.11) and fatigue (SMD = −0.09, 95% CI = −0.33, 0.15; *p* = 0.44). Significant overall effects were observed in favour of exercise-based interventions on aerobic fitness (SMD = 0.50, 95% CI = 0.15, 0.85; *p* < 0.01), upper-body strength (SMD = 0.34, 95% CI = 0.04, 0.63; *p* = 0.03), and lower-body strength (SMD = 0.54, 95% CI = 0.26, 0.82; *p* < 0.01). Results of subgroup analyses are shown in Appendix A. Intervention length had effects on fatigue (test for subgroup differences: χ^2^ = 7.09, df = 1, *p* < 0.01) and upper-body strength (test for subgroup differences: χ^2^ = 4.12, df = 1, *p* = 0.04). Interventions that were >12 weeks had larger effects on fatigue (SMD = 0.18, 95% CI = −0.07, 0.43) and upper-body strength (SMD = 0.72, 95% CI = 0.27, 0.18) than ≤12-week interventions (fatigue: SMD = −0.29, 95% CI = −0.54, −0.05; upper-body strength: SMD = 0.17, 95% CI = −0.11, 0.45). Furthermore, exercise mode had an effect on upper-body strength (test for subgroup differences: χ^2^ = 4.12, df = 1 (*p* = 0.04), *I*^2^ = 75.7%), with resistance exercise having a large effect (SMD = 0.72, 95% CI = 0.27, 1.18; *p* < 0.01), compared with no effect of mixed mode exercise (SMD = 0.17, 95% CI = −0.11, 0.45; *p* = 0.24). Exercise adherence, exercise mode, and intervention length had no other subgroup effects on the outcomes of interest.

### 3.5. Participant Adherence

The included quantitative articles describe the adherence of participants to the prescribed exercise intervention in multiple ways, including reporting the exercise adherence and reporting study attrition. All included quantitative studies reported the attrition rates, with any studies not reporting attrition or exercise adherence excluded from this review during article screening.

There were five studies that included a description of the adverse events that occurred as the result of their intervention, including musculoskeletal events [28,30,39,47,48], with fibula fractures also reported by Uth, et al. [47], who prescribed a group football intervention. In contrast, there were two studies that specified that no adverse events were reported by their participants [31,40]; however, the reporting of adverse events was not in the inclusion criteria for this review, with studies not being required to report them to be eligible.

Of the included studies, 14 (48%) reported the reasons for participant withdrawal [24,25,27,28,30,37,39,41,42,45,46,48,49,50], with commonly reported reasons including unrelated medical problems, ADT side effects, time constraints, lost contact, lack of interest, and individuals in the control group wanting to participate in exercise. While they reported the total numbers of participant withdrawals, there are several studies that did not include a description of the withdrawal reasons [31,32,34,38]. Although the reasons for withdrawal were not reported by Focht, et al. [29] and Uth, et al. [47], these were included in other articles for their studies [28,45,46], with no reasons provided for the study by Segal, et al. [44] and Courneya, et al. [26]. A single study reported no participant dropouts throughout their intervention, with this study intervention prescribing supplementation, a dietary program, and tailored exercise (regular aerobic training) advice for 6 months [40].

### 3.6. Qualitative Study Findings

The qualitative studies included in this review provided important information regarding the perspectives of patients with prostate cancer on the barriers and enablers of participating in exercise during their cancer experience (Table 2). The studies that met the inclusion criteria in this review presented both formal exercise programs and interventions, and self-driven unsupervised exercise programs. These studies revealed the importance of the social aspect of formal exercise programs completed in a group setting [23], reporting that some participants continued to exercise together following the completion of their study [43]. Participants also highlighted that they struggled with connecting with new peers who had not experienced prostate cancer or ADT once leaving the program [43], highlighting the clear need for support and connection for this population. The participants in the study also reported that the group setting provided motivation for exercise participation, particularly when ADT side effects were severe [43]. In contrast, those who did not experience a group program preferred to exercise alone due to worries of feeling judged about their physical abilities [33], including experiencing self-judgement as a barrier for participation at all [51].

Participants in supervised exercise programs identified the benefit of feedback on exercise progression by the supervisors [23], with further encouragement sometimes desired [43]. The skills learnt in the supervised sessions assisted with participant progression to unsupervised settings [35,43]; however, home-based exercise was reported to be less motivating [23]. Similarly, there was a hesitancy for participants to continue exercising in gym settings outside of formal programs due to not knowing if the new setting would be able to support their needs [23].

Exercise being acknowledged by the participants as a strategy to improve the side effects of ADT [33,43] was another theme identified in this review. With changes in body image also recognised as a common side effect of ADT and a barrier by participants [33,35], the benefits of exercise were perceived to outweigh any risks [36]. The benefit of exercise on improving treatment side effects was also identified as being a motivator for the continuation of exercise following formal exercise program completion [51]; however, severe ADT side effects were also acknowledged to be a barrier of exercise participation [33,43]. These statements highlight the important need for education for this population and support by the cancer care team; it also acknowledges that men with prostate cancer undergoing ADT need more programs (both individual and group-based) and support.

## 4. Discussion

The primary outcome of this systematic review was to determine factors that affect adherence to exercise programs in participants with prostate cancer receiving ADT. The current meta-analysis showed that adherence to exercise-based interventions improved aerobic fitness and upper- and lower-body strength, with no improvements observed overall for quality of life. Subgroup analysis showed that intervention length improved fatigue and upper-body strength with greater than 12-week interventions having a larger effect compared to less than 12 weeks. The mode of exercise had an impact on upper-body strength with improvements observed from resistance training, and no effect was seen from the mixed mode interventions. These results show that adhering to longer-term exercise interventions is important for this population; it also highlights that exercise alone may not be enough for these men to improve quality of life, and that multidisciplinary interventions with psychological support are needed.

The adherence to the prescribed exercise interventions was reported by several of the included studies [24,26,28,29,38,39,44,48], with the reported average adherence at 80.38% across these studies. Furthermore, the highest reported adherence was 94% adherence to supervised sessions [24], with 49% for home-based exercise being the lowest reported adherence [48]. Interestingly, Via, et al. [48] included both supervised and unsupervised exercise sessions in their study, with 65% adherence to supervised sessions and 49% reported for unsupervised sessions [48]. In contrast, the study by Villumsen, et al. [49] reported the duration of exercise completed by participants compared to the prescribed exercise duration, with 153.5 of the prescribed 180 min per week completed, highlighting that one blanket amount of exercise may not be achievable for each participant; however, individual improvements may still be observed.

The type of exercise prescribed and the exercise environment may be factors in determining possible adherence to exercise in this population. One of the included studies described the maintenance of self-directed exercise levels in the intervention group following the withdrawal of exercise supervision [38]. However, a reduction in physiological outcomes, such as cardiopulmonary fitness and fatigue was observed. Despite this, the participants of this group reported a greater quality of life and reduced cardiovascular events risk than controls, highlighting some of the possible advantages of exercise supervision in this population [38]. Further advantages include the benefits of expert feedback [23] and skill acquisition [33,43]. Similarly, group exercise within this population has been reported to be a motivating setting [43], in addition to providing a social opportunity [23]. These important studies emphasise the urgent need for support for men participating in an exercise program so that they can complete it successfully in a way that may motivate them (group or individually) to achieve reported benefits.

Some additional factors that have been reported to impact participant adherence in intervention groups, as described in this review, include study visits being too time intensive [37,45,48] and a loss of motivation or interest in the prescribed exercises [38], particularly for those described as tedious [42]. Interestingly, participants in several of the included control groups elected to discontinue their involvement due to being dissatisfied with group allocation and wanting to exercise [25,48].

### 4.1. Strengths and Limitations

To the best of our knowledge, this is the first study to investigate the adherence of patients with prostate cancer to exercise interventions. Strengths of this review include the use of PRISMA guidelines, the use of only RCTs for the meta-analyses, and the analysis of subgroups to identify effects of important intervention components. This review also explores the experiences of men receiving ADT to exercise qualitatively, providing a unique insight into some of the potential factors influencing their participation. This review is limited by the selection criteria, which specified that only studies that had a control group of usual care were to be included. This resulted in the exclusion of several articles that had exercise of a lesser intensity prescribed to their control group. Other limitations of this review include the lack of assessor blinding and small sample sizes in the included studies, and the limited number of studies with longer-term follow-up (e.g., >1 year).

### 4.2. Directions for Future Research

The issues with adherence discussed in this review will be taken into consideration in the design of a randomised controlled exercise intervention for men with prostate cancer receiving ADT. Future research should also aim to evaluate the longer-term effects of exercise in this population with long-term follow-up assessments (e.g., >1 year), including the effects of exercise on other important disease-related outcomes such as disease progression and survival. This will allow for the potential of greater intervention compliance and increased benefits experienced in this population.

## 5. Conclusions

Improvements were observed in aerobic fitness and muscle strength in this review, and adherence to exercise-based interventions was 80.38% overall. Exercise adherence had a positive impact on fatigue and muscle strength. Programs greater than 12 weeks showed greater improvements in fatigue, muscle strength, and adherence to the exercise programs. Participants reported both in favour of group and individual exercise programs to stay motivated. Interestingly, in this review, adherence to exercise did not impact quality of life, highlighting the need for exercise professionals to carefully monitor and provide support and referrals for these men.

## Figures and Tables

**Figure 1 cancers-14-02452-f001:**
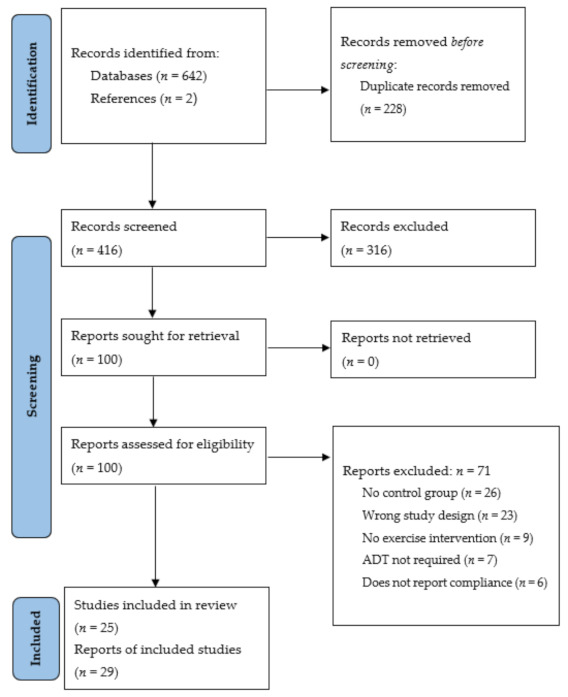
Search strategy and article selection process according to the Preferred Reporting Items for Systematic Reviews and Meta-Analyses (PRISMA) guidelines [16].

**Figure 2 cancers-14-02452-f002:**
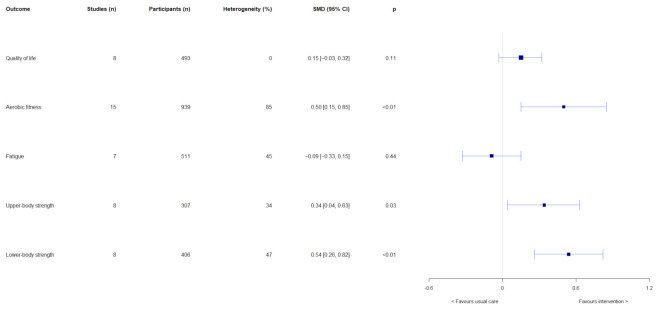
Results of meta-analyses on the overall effects on quality of life, aerobic fitness, fatigue, upper-body strength, and lower-body strength.

**Table 1 cancers-14-02452-t001:** Quantitative Studies Summary (*n* = 23 articles from *n* = 19 studies).

Study; Country; Setting	Participants	Intervention	Duration	Adherence
Bourke et al., 2014; United Kingdom; rehabilitation centre [24]	Treatment: ADT ≥6 months Participants: Intervention: 71 ± 6 years (*n* = 50) Control: 71 ± 8 years (*n* = 50)	Intervention: tapered (supervised) exercise and dietary intervention. Supervised: aerobic (30 min, 55–75% of age predicted HR_max_ or 11–13 RPE; cycle and rowing ergometers, and treadmill); resistance (progressive 2–4 sets and 8–12 repetitions beginning at 60% of 1 RM) exercise; dietary advice and behaviour change support. Self-directed exercise (walking, cycling, and gym exercise using skills learnt in supervised sessions, such as RPE) Control: usual care	Weeks 1–6: 2 supervised exercise sessions/week and at least 1 self-directed independent exercise session Weeks 7–12: 1 supervised exercise session/week and at least 2 self-directed independent exercise sessions	Intervention: − 86% retention − Lost to follow-up (before 12 weeks); unrelated medical problems (*n* = 3), accident at home (*n* = 1), developed atrial fibrillation (*n* = 1), increased family commitments (*n* = 2) − Dropped out (after 12 weeks); accident at home (*n* = 1), unrelated medical problems (*n* = 3), no response (*n* = 4) − Adherence was 94% for supervised and 82% for independent exercise sessions Control: − 84% retention − Lost to follow-up (before 12 weeks); no response (*n* = 5), unrelated death (*n* = 1), developed medical problems (*n* = 2) − Dropped out (after 12 weeks) total *n* = 9; developed unrelated medical problems (*n* = 5), no response (*n* = 4)
Cormie et al., 2015; Australia; multicentre [25]	Treatment: commencing leuprorelin acetate for >3 months Participants: Intervention: 69.6 ± 6.5 years (*n* = 32) Control: 67.1 ± 7.5 years (*n* = 31)	Intervention: progressive moderate–high intensity aerobic (treadmill, stationary ergometer, cross trainer; target intensity of 70–85% HR_max_) and resistance (major muscle groups; intensity of 6–12 RM for 1–4 sets) exercises; recommended 150 min moderate intensity aerobic exercise Control: usual care	60-min sessions twice weekly for 3 months	Intervention: − 97% retention − ADT side effects (*n* = 1) Control: − 77% retention − Wanted to exercise (*n* = 4), distance (*n* = 2), time constraints (*n* = 1)
Culos-Reed et al., 2010; Canada; fitness centre [27]	Treatment: ADT ≥6 months Participants: Intervention: 67.2 ± 8.8 years (*n* = 53) Control: 68.0 ± 8.4 years (*n* = 47)	Intervention: home-based and weekly group sessions ( walking, stretching, and light resistance exercise) Control: usual care	16 weeks Home-based: recommended 3–5 times weekly Supervised: 1.5-h sessions (1 h activity, 30 min educational)	Intervention: − 79% retention − Lost to follow-up (*n* = 2), voluntarily withdrew (*n* = 3), medical (*n* = 5), unknown (*n* = 1) Control: − 51% retention − Lost to follow-up (*n* = 11), voluntarily withdrew (*n* = 6), medical (*n* = 3), unknown (*n* = 3)
Focht et al., 2018; USA; multicentre [28] Focht et al., 2019; USA; multicentre [29]	Treatment: ADT Participants: Intervention: 69.4 ± 9.0 years (*n* = 16) Control: 64.5 ± 8.6 years (*n* = 16)	Intervention: Supervised and tailored progressive resistance (3 sets at 8–12 RM for 9 exercises) and aerobic (10–20 min 3–4 RPE on aerobic machines) exercise; group-mediated cognitive behavioural counselling, dietary counselling, and education Control: usual care	Intervention: 12 weeks, twice a week, 1 h; diet: once a week, one hour, group setting, 8 weeks, followed by bi-weekly phone calls weeks 9–12	Intervention: − 88% retention − Adverse events: no serious events. Exercise-related nausea (*n* = 1), musculoskeletal pain (*n* = 1) − Adherence to supervised exercise sessions was 88%, dietary sessions was 84% Control: − 69% retention 2-month follow-up (reported for both groups): − Missed/lost contact (*n* = 4), dropped out (*n* = 6) 3-month follow-up (reported for both groups): − Missed/lost contact (*n* = 1), dropped out (*n* = 6)
Freedland et al., 2019; USA; setting NR [30]	Treatment: commencing ADT (LHRH agonist, LHRH antagonist, or orchiectomy) Participants: Intervention: 66 (61–76) years (*n* = 20) Control: 66 (56–70) years (*n* = 22)	Intervention: carbohydrate intake ≤20 g/day, and walking ≥30 min/day Control: usual care	6 months ≥30 min walking/day for ≥5 days/week	Intervention: − 70% retention − Ineligible (*n* = 1), lost to follow-up at 3 months (*n* = 2), withdrew at 6 months due to diet, schedule, or work (*n* = 3), excluded from analysis (incomplete data) (*n* = 3) − Adverse events: fatigue, constipation, and headaches Control: − 91% retention − Withdrew after allocation (*n* = 1), lost to follow-up at 3 months (*n* = 2), excluded from analysis (incomplete data) (*n* = 2)
Galvao et al., 2010; Australia; setting NR [31]	Treatment: AST ≥2 months Participants: Intervention: 69.5 ± 7.3 years (*n* = 29) Control: 70.1 ± 7.3 years (*n* = 28)	Intervention: combined progressive resistance (exercises using major muscle groups; 12–6 RM for 2–4 sets; general flexibility exercises) and aerobic (15–20 min cycling and walking/jogging at 65–80% HR_max_ at 11–13 RPE) training Control: usual care	Twice a week for 12 weeks	Intervention: − 97% retention − Discontinued (*n* = 1) − No adverse events reported Control: − 96% retention − Lost to follow-up (*n* = 1)
Gazova et al., 2019; Slovak Republic; university [32]	Treatment: ADT for 24–36 weeks Participants: Intervention: 69.21 ± 5.8 years (*n* = 15) Control: 70.69 ± 7.5 years (*n* = 8)	Intervention: progressive resistance training: Month 1: 30% resistance, 2 series, 4 exercises, 10–15 reps. Month 2 and 3: 90–100% resistance, 2 series, 5 exercises, 10–12 reps. Month 4: 90–100% resistance, 3 series, 5 exercises, 10–15 reps Control: usual care	3 times/week for 16 weeks	Reported for both groups: − 72% retention − Discontinued (*n* = 9)
Gilbert et al., 2016; United Kingdom; setting NR [34]	Treatment: long-term ADT ≥6 months Participants: Intervention: 70.1 ± 5.3 years (*n* = 25) Control: 70.4 ± 9.2 (*n* = 25)	Intervention: combined supervised aerobic (30 min at 55–75% predicted age, predicted HR_max_, or 11–13 RPE scale using cycling, rowing, or treadmill machines), resistance (2–4 sets of 8–12 reps beginning at an intensity of 60% of 1 RM) and balance exercises. Instructions provided for 30 min at home exercises Healthy eating seminars provided dietary advice Control: usual care	Three 1-h sessions/week for 12 weeks Healthy eating seminars held every 2 weeks	Intervention: − 88% retention − 6 weeks: lost to follow-up (*n* = 1) − 12 weeks: lost to follow-up (*n* = 2) − 24 weeks: lost to follow-up (*n* = 1), death (*n* = 1) Control: − 80% retention − 6 weeks: lost to follow-up (*n* = 3), death (*n* = 1) − 12 weeks: lost to follow-up (*n* = 1)
Lam et al., 2020; Australia; setting NR [37]	Treatment: GnRH analogues Participants: Intervention: 69.3 ± 2.3 years (*n* = 13) Control: 71.8 ± 1.8 years (*n* = 12)	Intervention: progressive individualised resistance training (8–10 exercises targeting major muscle groups using dumbbells or body weight; 3 sets of 8–12 RM) Control: usual care	12 months, 3 times a week	Intervention: − 77% retention − 6 months: study visits too time intensive (*n* = 1) − 12 months: housing relocation (*n* = 1), discontinued ADT (*n* = 1) Control: − 83% retention − 6 months: did not attend follow-up; however, continued participation (*n* = 1) − 12 months: housing relocation (*n* = 1), discontinued ADT (*n* = 1)
Ndjavera et al., 2020; United Kingdom; university hospital [38]	Treatment: commencing LHRH agonist with or without RT Participants: Intervention: 71.4 ± 5.4 years (*n* = 24) Control: 72.5 ± 4.2 years (*n* = 26)	Intervention: supervised aerobic interval (cycle ergometer; 11–15 RPE) and resistance training (targeting major muscle groups; 2–4 sets of 10 repetitions at 11–15 RPE). Patients also advised to engage in home-based physical activity and instructed to continue exercising following 12 weeks of supervision Control: usual care	2 × 60 min sessions per week for 12 weeks Home-based: recommended 30-min 3 times a week	Retention: − Intervention 92%, control 77% − All patients in exercise group completed at least 17/24 supervised sessions (≥70%) 3 months: − Lack of motivation/interest (*n* = 2 participants in each group) 6 months: − Missed assessments (*n* = 13 across both groups)
Nilsen et al., 2015; Norway; setting NR [39]	Treatment: GnRH analogue and RT Participants: Intervention: 66 ± 6.6 years (*n* = 28) Control: 66 ± 5 years (*n* = 30)	Intervention: progressive strength training program; 9 exercises of the major muscle groups; Mondays 1–3 sets at 10 RM, Wednesday 10 repetitions at 80–90% of 10 RM in 2–3 sets, Friday 2–3 sets at 6 RM Control: usual care	3 sessions per week for 16 weeks	Intervention: − 79% retention − Pain (knee *n* = 2 and back *n* = 1), accident not related to the study (*n* = 2), hospitalised not related to the study (*n* = 1) − Completed 88% of the training sessions for lower-body exercises (64–98%), 84% for upper-body exercises (69–98%) Control: − 90% retention − Hospitalised due to infection (*n* = 2), knee pain (*n* = 1)
Nobes et al., 2012; UK; setting NR [40]	Treatment: ADT Participants: Intervention: 70.5 (58–80) years (*n* = 20) Control: 69.5 (56–84) years (*n* = 20)	Intervention: patients provided with metformin (commenced at 850 mg daily, increased to 850 mg twice daily after 2 weeks), dietary (low glycaemic index diet), and tailored exercise (regular aerobic exercise) advice from the onset of ADT administration Control: usual care	6 months	Retention: − 100% in both groups − No participant dropouts, no adverse effects reported
O’Neill et al., 2015; Northern Ireland; multicentre [41]	Treatment: LHRH agonist Participants: Intervention: 69.7 ± 6.8 years (*n* = 45) Control: 69.9 ± 7.0 (*n* = 45)	Intervention: Pedometer provided for tracking walking; dietary guide provided based on usual diet and UK recommendations Control: usual care	Recommended 30 min walking 5 times a week for 6 months; 7-day food diary at endpoint	Intervention: − 96% retention − Disease progression (*n* = 1), carer duties (*n* = 1) Control: − 96% retention − Diagnosis of lung cancer (*n* = 1), accidental death (*n* = 1)
Sajid et al., 2016; USA; multicentre [42]	Treatment: ADT Participants: Intervention 1 (Wii): 77.5 ± 6.7 years (*n* = 8) Intervention 2 (EXCAP): 75.7 ± 9.5 years (*n* = 6) Control: 71.8 ± 5.0 years (*n* = 5)	Intervention 1; Wii-Fit: individually tailored, provided with Wii-Fit technology, instruction, and pedometer. Intervention 2; EXCAP: provided with a pedometer and resistance bands. Aerobic (walking program) and resistance (band exercise) Control: usual care	6 weeks Intervention 1: as prescribed by exercise physiologist Intervention 2: walking and resistance program 5 days/week	Intervention 1: − 63% retention − Misplaced equipment (*n* = 1), loss of interest in the exercises (*n* = 2) Intervention 2: − 83% retention − Exercises were tedious (*n* = 1) Control: − 60% retention − Completing diaries was cumbersome (*n* = 2)
Segal et al., 2003; Canada; multicentre [44] Courneya et al., 2004; Canada; Cancer Centre [26]	Treatment: ADT Participants: Intervention: 68.2 ± 7.9 years (*n* = 82) Control: 67.7 ± 7.5 years (*n* = 73)	Intervention: 9 strength training exercises at 60–70% of 1 RM, increasing weight by 5lb when 12 repetitions was completed Control: usual care	12 weeks, 3 times per week	Intervention: − 90% retention − Discontinued (*n* = 8) − Attendance to exercise sessions averaged 79% Control: − 84% retention − Discontinued (*n* = 12)
Uth et al., 2014; Denmark; Multicentre [45] Uth et al., 2016a; Denmark; Multicentre [46] Uth et al., 2016b; Denmark; Multicentre [47]	Treatment: ADT ≥6 months Participants: Intervention: 67.1 ± 7.1 years (*n* = 29) Control: 66.5 ± 4.9 years (*n* = 28)	Intervention: football: warm-up exercises (running, dribbling, passing, shooting, balance, and muscle strength) and small-sided games Control: usual care	12 weeks 2–3 times weekly; warm-up = 15 min, weeks 1–4 2 session, 2 × 15 min games; weeks 5–8 2 sessions 3 × 15 min games; weeks 9–12 3 sessions 3 × 15 min games; weeks 13–32: 2 weekly sessions, 1 h duration	Intervention: − 12 weeks: 90% retention − No time (*n* = 1), disliked football (*n* = 1), muscle strain (*n* = 1) − 32 weeks: 72% retention − Neuropathy (*n* = 1), deteriorating health (*n* = 3), not motivated (*n* = 1); *n* = 5 − Sustained musculoskeletal injuries: fibula fracture (*n* = 2), muscle or tendon injuries (*n* = 3), with 3 returning to participation during study period Control: − 12 weeks: 82% retention − No ADT (*n* = 2), chemotherapy (*n* = 1), treatment abroad (*n* = 1), not motivated (*n* = 1) − 32 weeks: 71% retention − No time (*n* = 1), unable to contact (*n* = 1), not motivated (*n* = 1)
Via et al., 2021; Australia; multicentre [48]	Treatment: ADT ≥12 weeks Participants: Intervention: 71.4 ± 5.9 years (*n* = 34) Control: 71.1 ± 6.6 (*n* = 36)	Intervention: gym-based (aerobic warm-up, progressive resistance exercises (2 sets, 8–12 repetitions, moderate to hard intensity), weight-bearing impact exercises (3 sets, 10–20 repetitions), balance exercises (2 sets, 30–60 s), core stability (2 sets, 10–15 repetitions)); home-based (body weight and resistance bands); multinutrient supplement (whey protein, calcium, vitamin D enriched drink, and vitamin D tablet) Control: usual care	12 months Gym-based: 60 min, 2 sessions weekly, both supervised in first 6 months, 1 session supervised in second 6 months Home-based: 20–60 min, 1 session weekly	Intervention: − 91% retention − 6 months: health issues (*n* = 1) − 12 months: health issues (*n* = 2); took supplement and did not complete exercise due to time constraints (*n* = 1), discontinued training (health issues (*n* = 3), lack of time (*n* = 1), personal reasons (*n* = 1) − Mean exercise adherence 56% ± 30% (supervised 65% ± 25%, unsupervised 49% ± 38%), mean supplement adherence 77% ± 30% − Minor musculoskeletal events reported (41%), participants (*n* = 3) stopped taking supplement due to adverse gastrointestinal complaints Control: − 81% retention − Baseline: dissatisfied with group allocation (*n* = 1) − 6 months: deceased (*n* = 1), disinterested (*n* = 1), health issues (*n* = 2) − 12 months: health issues (*n* = 2)
Villumsen et al., 2019; Denmark; multicentre [49]	Treatment: ADT ≥3 months Participants: Intervention: 67.6 ± 4.6 years (*n* = 23) Control: 69.8 ± 4.4 years (*n* = 23)	Intervention: home-based aerobic and strength exercise using free weights Control: usual care	3 × 1 h/week, 12 weeks	Intervention: − 91% retention − Withdrawal of consent (*n* = 1), non-cardiac-related chest pain (*n* = 1) − Protocoled exercise duration = 180 min/week; average recorded exercise duration = 153.5 min/week Control: − 87% retention − Allocation: withdrawal of consent (*n* = 1) − Follow-up: withdrawal of consent (*n* = 1), excluded (*n* = 2)
Wall et al., 2017; Australia; university clinic [50]	Treatment: ADT ≥2 months Participants: Intervention: 69.1 ± 9.4 years (*n* = 50) Control: 69.1 ± 8.4 years (*n* = 47)	Intervention: Aerobic: 70–90% participant heart rate using aerobic machines; progressive resistance: 6 exercises that targeted major muscle groups Control: usual care	6-month intervention; twice weekly 60-min clinic sessions	Intervention: − 86% retention − Health (*n* = 1), injury (*n* = 1), disinterest (*n* = 1), ineligible (bone metastases) (*n* = 2), other (*n* = 2) Control: − 70% retention − Health (*n* = 1), injury (*n* = 2), disinterest (*n* = 4), moved away (*n* = 1), deceased (*n* = 1), uncontactable (*n* = 1), ineligible (bone metastases) (*n* = 1), personal issues (*n* = 1), other (*n* = 2)

Note: ADT—androgen deprivation therapy; RT—radiotherapy; NR—not reported; HR_max_—maximum heart rate; RPE—rate of perceived exertion; RM—repetition maximum; GnRH—gonadotropin-releasing hormone; LHRH—luteinising hormone releasing hormone; AST—androgen suppression therapy, EXCAP—home-based aerobic and progressive resistance exercise program.

**Table 2 cancers-14-02452-t002:** Qualitative Studies Summary (*n* = 6 studies).

Study; Country; Setting	Participants	Study Design	Themes
Bourke et al., 2012; United Kingdom; university [23]	PCa patients receiving AST for at least 6 months, enrolled in an intervention (tapered supervised exercise program, nutrition advice pack, and healthy eating seminars) *n* = 12 participants	Focus groups (*n* = 3 in total)	Process themes: − Motivations for taking part in the study − Views about the supervised group design of the program − Perceived benefits of the social interaction within the group-based program − Views on home-based section of the exercise program − Perceived benefits from the diet aspect of the program − Factors that could affect future program participation − Impact on exercise behaviour after the intervention Outcome themes: − Disease recurrence − Communication with healthcare professionals − Benefits and drawbacks from taking part in the intervention
Gentili et al., 2019; United Kingdom; university [33]	PCa patients who had received ADT at some point, and were not prevented from exercising (67.9 ± 9.99 years, *n* = 22)	Individual semi-structured interviews over the phone (*n* = 13) and face-to-face (*n* = 9)	− Body image issues such as body feminisation issues − Compromising exercise and side effects: between compensation and barriers − Psychological implications of exercise: between empowerment and fear of evaluation
Hamilton et al., 2015; Australia; university [35]	PCa patients receiving ADT for ≤ 12 months were randomised into exercise (63.1 ± 3.8 years, *n* = 11, involvement 4.3 ± 2.4 months) and a usual care (60.3 ± 6.9 years, *n* = 7) group	Semi-structured interviews	− Concerns about sexual health − Coping with sexual health concerns − Exercise to combat sexual health concerns
Keogh et al., 2013; Australia; recruitment from urologists [36]	Fourteen men with prostate cancer; non-ADT (65.0 ± 6.5 years, *n* = 8) and ADT (65.8 ± 11.3 years, *n* = 6) participants	Semi-structured focus groups	− Perceived quality of life post-diagnosis − Physical activity engagement post-diagnosis − Perceived benefits of physical activity − Perceived risks of physical activity
Schmidt et al., 2019 Denmark; urology clinic (exercise programme), hospital (exercise) [43]	PCa patients receiving ADT Exercise programme: twice a week, 12-week supervised individual resistance (exercise machines) and aerobic exercise programme in groups of 10–15 men, with week 12 exercise being completed at a local fitness centre Interviews: included 29 (median age 71 (interquartile range 67–74) years) participants who had completed the exercise programme at least 2–3 months prior, and therefore had experienced the transition to unsupervised, community-based exercise	Semi-structured, open-ended focus groups (*n* = 5, up to 7 participants each)	− Development and practice of new skills − Establishing social relationships − Familiarising with bodily well-being
Wright-St Clair et al., 2014; New Zealand; interviewed from participant’s homes [51]	3 participants, (74–88 years) with prostate cancer using ADT continuously for at least 12 months and regularly exercising for at least 6 months (between 2 and 5 years)	Individual semi-structured interviews	− Getting started − Having a routine − Being with music

Note: PCa—prostate cancer; AST—androgen suppression therapy; ADT—androgen deprivation therapy.

**Table 3 cancers-14-02452-t003:** Assessment of quality appraisal in the included studies.

Randomised Controlled Trials	Item Number of Check List						
	S1.	S2.	1.1.	1.2.	1.3.	1.4.	1.5.
Bourke et al., 2014 [24]	Y	Y	Y	Y	Y	Y	Y
Cormie et al., 2015 [25]	Y	Y	Y	Y	Y	U	Y
Courneya et al., 2004 [26]	Y	Y	Y	Y	Y	U	Y
Culos-Reed et al., 2010 [27]	Y	Y	U	Y	Y	U	Y
Focht et al., 2018 [28]	Y	Y	U	Y	Y	Y	Y
Focht et al., 2019 [29]	Y	Y	Y	Y	Y	Y	Y
Freedland et al., 2019 [30]	Y	Y	Y	Y	Y	U	Y
Galvao et al., 2010 [31]	Y	Y	Y	Y	Y	U	U
Gazova et al., 2019 [32]	Y	Y	U	Y	Y	U	U
Gilbert et al., 2016 [34]	Y	Y	Y	Y	Y	U	Y
Lam et al., 2020 [37]	Y	Y	Y	Y	Y	U	U
Ndjavera et al., 2020 [38]	Y	Y	Y	Y	Y	Y	Y
Nilsen et al., 2015 [39]	Y	Y	Y	Y	Y	U	Y
Nobes et al., 2012 [40]	Y	Y	Y	Y	Y	U	Y
O’Neill et al., 2015 [41]	Y	Y	Y	Y	Y	N	Y
Sajid et al., 2016 [42]	Y	Y	U	Y	Y	Y	Y
Segal et al., 2003 [44]	Y	Y	Y	Y	Y	Y	Y
Uth et al., 2014 [45]	Y	Y	Y	Y	Y	U	U
Uth et al., 2016a [46]	Y	Y	Y	Y	Y	U	Y
Uth et al., 2016b [47]	Y	Y	Y	Y	Y	U	Y
Via et al., 2021 [48]	Y	Y	Y	Y	Y	U	Y
Villumsen et al., 2019 [49]	Y	Y	Y	Y	Y	Y	Y
Wall et al., 2017 [50]	Y	Y	Y	Y	Y	U	Y
**Qualitative Studies**	**Item Number of Check List**						
	**S1.**	**S2.**	**2.1.**	**2.2.**	**2.3.**	**2.4.**	**2.5.**
Bourke et al., 2012 [23]	Y	Y	Y	Y	Y	Y	Y
Gentili et al., 2019 [33]	Y	Y	U	Y	Y	Y	Y
Hamilton et al., 2015 [35]	Y	Y	Y	Y	Y	Y	Y
Keogh et al., 2013 [36]	Y	Y	Y	Y	Y	Y	Y
Schmidt et al., 2019 [43]	Y	Y	Y	Y	Y	Y	Y
Wright-St Clair et al., 2014 [51]	Y	Y	Y	Y	Y	Y	Y

Item number check list key *: S1. Are there clear research questions? S2. Do the collected data allow to address the research questions? 1.1. Is randomisation appropriately performed? 1.2. Are the groups comparable at baseline? 1.3. Are there complete outcome data? 1.4. Are outcome assessors blinded to the intervention provided? 1.5. Did the participants adhere to the assigned intervention? 2.1. Is the qualitative approach appropriate to answer the research question? 2.2. Are the qualitative data collection methods adequate to address the research question? 2.3. Are the findings adequately derived from the data? 2.4. Is the interpretation of results sufficiently substantiated by data? 2.5. Is there coherence between qualitative data sources, collection, analysis, and interpretation? * Three levels of assessment quality scores. Y = Yes; U = Unclear; N = No.

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
