# Peer review of "Exercise Adherence in Men with Prostate Cancer Undergoing Androgen Deprivation Therapy: A Systematic Review and Meta-Analysis"

_cancers, 2022, doi:10.3390/cancers14102452_

Round 1

Reviewer 1 Report

Prostate cancer (PCa) is the second most common neoplasm in men and the fifth cause of death, worldwide. Pharmacological strategies include hormonotherapy.

Hormonotherapy, also known as androgen deprivation therapy (ADT), has the purpose to abolish gonadic testosterone levels and prostatic tumor stimulation to growth.

ADT is used as adjuvant or rescue therapy in metastatic pN+ patients, high-risk localized and locally advanced tumors in association with radiotherapy (RT), or in patients with reduced life expectancy.  

ADT has several side effects too such as hot flashes, asthenia, gynecomastia, erectile dysfunction, loss of libido, loss of muscle mass as well as the appearance of metabolic syndrome, increased risk of death from cardiovascular causes and hepatotoxicity.

This study aims to investigate exercise intervention adherence of patients receiving ADT, while identifying some of the effects of exercise on some physiological outcomes.

COMMENTS TO AUTHORS:

The authors should be congratulated for the interesting topic discussed.

Despite the countless successes in the clinical field when it comes to approaching to PCa, we have not to forget comorbidities that drug therapy such as ADT, although correct, can add to the patient, worsening their quality of life.

Having the chance to administer a counter-therapy in these cases could be a great solution for both patient and specialist, even thou randomized controlled studies about it are still lacking.

I believe that the study has sufficient merit to be considered for publication, although major revisions are required.

  1. Methods and methodology are robust.
  2. Results, conclusions, and limitations are well presented.
  3. Referred to lines 57 to 62: the authors should give more detailed information on the side-effects of antiandrogenic therapy, above all their different impacts on the cardiovascular system. A lecture on this interesting study (https://doi.org/10.3389/fendo.2021.695170) could enhance the scientific value of the paper.

Despite the countless successes in the clinical field when it comes to approaching to PCa, we have not to forget comorbidities that drug therapy such as ADT, although correct, can add to the patient, worsening their quality of life.

Having the chance to administer a counter-therapy in these cases could be a great solution for both patient and specialist, even thou randomized controlled studies are still lacking.

I suggest that studies like this could lay the foundations for more in-depth research into the treatment of ADT side effects.

Author Response

Reviewer 1:

The authors should be congratulated for the interesting topic discussed.

We would like to thank Reviewer 1 for their time taken to provide suggestions to improve our manuscript.

Despite the countless successes in the clinical field when it comes to approaching to PCa, we have not to forget comorbidities that drug therapy such as ADT, although correct, can add to the patient, worsening their quality of life.

Having the chance to administer a counter-therapy in these cases could be a great solution for both patient and specialist, even thou randomized controlled studies about it are still lacking.

I believe that the study has sufficient merit to be considered for publication, although major revisions are required.

  1. Methods and methodology are robust.
  2. Results, conclusions, and limitations are well presented.
  3. Referred to lines 57 to 62: the authors should give more detailed information on the side-effects of antiandrogenic therapy, above all their different impacts on the cardiovascular system. A lecture on this interesting study (https://doi.org/10.3389/fendo.2021.695170) could enhance the scientific value of the paper.

We would like to thank the reviewer for the suggestion to provide more detailed information on the side effects of the antiandrogenic therapy. We have included the reference suggested by the reviewer to improve the evidence for the association between antiandrogenic therapy and cardiovascular disease risk. Please see Lines 58-61:

These side effects include the significant risk of the development of obesity, diabetes mellitus, and cardiovascular disease [6], with an inverse association observed between serum testosterone levels and cardiovascular disease risk [7].

Despite the countless successes in the clinical field when it comes to approaching to PCa, we have not to forget comorbidities that drug therapy such as ADT, although correct, can add to the patient, worsening their quality of life.

Having the chance to administer a counter-therapy in these cases could be a great solution for both patient and specialist, even thou randomized controlled studies are still lacking.

I suggest that studies like this could lay the foundations for more in-depth research into the treatment of ADT side effects.

Reviewer 2 Report

Thank you to the Authors for the interesting paper

minor comment:

in the abstract please reword the following sentence: "Articles (n=64) articles were identified, with 29 (n=23 quantitative; n=6 qualitative) articles from 25 studies included"

in the conclusion section: please delete the following sentence "This section is mandatory, with one or two paragraphs to end the main text."

Author Response

In the abstract please reword the following sentence: "Articles (n=64) articles were identified, with 29 (n=23 quantitative; n=6 qualitative) articles from 25 studies included"

This sentence in the abstract has been updated to remove the duplicate “articles”, and now states in Lines 34-35:

In total, 644 articles were identified, with 29 (n=23 quantitative; n=6 qualitative) articles from 25 studies included.

In the conclusion section: please delete the following sentence "This section is mandatory, with one or two paragraphs to end the main text."

We would like to thank the reviewer for their attention to this editorial error. This sentence has been removed from the manuscript.

Round 2

Reviewer 1 Report

The authors answered all comments and suggestions.

Author Response

We would like to thank Reviewer 1 for their time taken to provide us with this feedback.